# Ubiquitination of Intramitochondrial Proteins: Implications for Metabolic Adaptability

**DOI:** 10.3390/biom10111559

**Published:** 2020-11-16

**Authors:** Prasad Sulkshane, Jonathan Ram, Michael H Glickman

**Affiliations:** The Faculty of Biology, Technion Israel Institute of Technology, Haifa 32000, Israel; jonathan.ram@campus.technion.ac.il

**Keywords:** mitochondria, ubiquitin, proteasome, mitophagy, autophagy, proteolysis, protein import, protein quality control, metabolism

## Abstract

Mitochondria are constantly subjected to stressful conditions due to their unique physiology and organization. The resulting damage leads to mitochondrial dysfunction, which underlies many pathophysiological conditions. Hence, constant surveillance is required to closely monitor mitochondrial health for sound maintenance of cellular metabolism and thus, for viability. In addition to internal mitochondrial chaperones and proteases, mitochondrial health is also governed by host cell protein quality control systems. The ubiquitin-proteasome system (UPS) and autophagy constitute the main pathways for removal of damaged or superfluous proteins in the cytosol, nucleus, and from certain organelles such as the Endoplasmic Reticulum (ER) and mitochondria. Although stress-induced ubiquitin-dependent degradation of mitochondrial outer membrane proteins has been widely studied, mechanisms of intramitochondrial protein ubiquitination has remained largely elusive due to the predominantly cytosolic nature of UPS components, separated from internal mitochondrial proteins by a double membrane. However, recent research has illuminated examples of intramitochondrial protein ubiquitination pathways and highlighted their importance under basal and stressful conditions. Owing to the dependence of mitochondria on the error-prone process of protein import from the cytosol, it is imperative that the cell eliminate any accumulated proteins in the event of mitochondrial protein import deficiency. Apparently, a significant portion of this activity involves ubiquitination in one way or another. In the present review article, following a brief introduction to mitochondrial protein quality control mechanisms, we discuss our recent understanding of intramitochondrial protein ubiquitination, its importance for basal function of mitochondria, metabolic implications, and possible therapeutic applications.

## 1. Introduction—Modes of Mitochondrial Protein Quality Control

Over 60 years ago, pioneering research cemented the identity of mitochondria as the “central powerhouse” of the cell. Over the subsequent decades, mitochondria were additionally recognized as performing diverse functions critical for cellular homeostasis beyond cellular bioenergetics. Some of these functions include synthesis of important metabolites (such as fatty acids, lipids, and quinones), regulation of metabolism, cell cycle, diverse signaling cascades, development, and neuronal functions through mitochondrial network dynamics, and regulation of apoptotic cell death [1]. Owing to the involvement of mitochondria in myriad cellular processes, their dysfunction has been implicated in several human pathophysiological conditions including neurodegenerative disorders [2,3], ageing [4,5], cardiovascular diseases [6], and cancer [7,8]. The causes of mitochondrial dysfunction under physiological conditions are diverse and remain to be properly understood. However, due to their unique physiological function and architecture it is clear that mitochondria are continuously subjected to at least two biochemical assaults: free radicals and unfolded proteins. First, being centers of oxidative phosphorylation (OXPHOS) and primary consumers of molecular oxygen (O_2_), mitochondria are an inadvertent source of reactive oxygen species (ROS) [9,10,11]. ROS thus generated further damage other mitochondria in a feed-forward manner causing a cascade, which if not contained by cellular antioxidant defenses can be catastrophic for the cell. Second, due to their unique architecture, mitochondria depend on a complex biogenesis process. As a majority of mitochondrial proteins are encoded by the nuclear genome, they are translated by cytosolic ribosomes and require efficient import into mitochondria [12,13]. Essentially, mitochondria are inundated with a continuous flux of unfolded extended polypeptides that must be properly localized to inner mitochondrial compartments, protected from aggregation, refolded, and incorporated into their quaternary structures. Moreover, at least about 13 mitochondrial proteins are encoded by the mitochondrial DNA and synthesized within the organelle itself. These proteins need to be efficiently assembled together with the imported proteins to form functional complexes [14]. Thus, mitochondrial biogenesis needs a high degree of coordination between protein sorting, import, folding, and assembly which makes it an error-prone process [15]. Mitochondrial quality surveillance mechanisms are therefore indispensable for ensuring sound maintenance of mitochondrial proteostasis, and preservation of mitochondrial integrity and their healthy pool in the cell [16].

Mitochondria have evolved multiple protein quality control mechanisms to protect proteins during their import, and protect mitochondria from damage when the process fails. Once synthesized in the cytosol, mitochondria-destined polypeptides are maintained in an unfolded state to aid in their translocation across the two mitochondrial membranes [17]. In this unfolded conformation, the nascent polypeptides are prone to misfolding or aggregation due to their inherent stretches of hydrophobic amino acids. Cytosolic chaperones such as HSP70 and HSP90 prevent these polypeptides from misfolding by shielding hydrophobic segments from the aqueous cytosol, and target them to the translocase in the outer membrane of the mitochondria (TOM complex) [18]. Next, the mitochondrial version of HSP70 takes over in the matrix and assists in the transmembrane transport of the precursor proteins by latching onto the mitochondrial targeting presequence as it traverses the translocase at the inner membrane of mitochondria (TIM23 complex) and exerting a mechanical pulling force [12,13,19]. This translocation process across the two mitochondrial membranes is facilitated by transmembrane potential across the inner membrane (ΔΨ; negative inside). Pulling a positively charged targeting sequence on the incoming polypeptide towards the negatively charged, matrix side, of the inner membrane, may also drive active unfolding of precursor proteins with short positively charged sequences thereby promoting translocation. Most proteins that engage the mitochondrial HSP70 are sorted into the mitochondrial matrix; in this context HSP70 facilitate protein import both at the TIM23 channel and in the matrix. The mitochondrial chaperone HSP60 together with HSP70 play an additional important role in (re)folding of the polypeptide chains and in intramitochondrial sorting of the imported proteins to mitochondrial subcompartments, including the intermembrane space and inner membrane [20].

Mitochondrial proteases also serve a critical role in the mitochondrial protein quality control. When unfolded proteins cannot be refolded back to their native conformation even with assistance from the chaperones, they need to be degraded. For this purpose, mitochondria are equipped with a unique set of proteases dedicated to specific subcompartments. Members of the Lon AAA^+^-protease fusion family reside in the matrix and degrade denatured or oxidized proteins [21]. Working in parallel to Lon, ClpP is another matrix-resident ATP-dependent protease which functions together with a AAA^+^ chaperone protein, ClpX [22]. For quality control of inner mitochondrial membrane (IMM) proteins, two additional AAA^+^ metalloproteases anchored within IMM are oriented opposite to each other. The catalytic domain of the m-AAA protease faces towards the matrix and performs its proteolytic function in the matrix, whereas the i-AAA protease faces the intermembrane space (IMS) and exhibits its activity there [23]. All these ATP-dependent proteases select their substrates within mitochondria, and release their peptide products into the mitochondria milieu (matrix or IMS), although some experiments identified peptides exported from mitochondria to the cytosol [23]. Apart from the i-AAA protease, IMS is equipped with a serine protease HtrA2/Omi, which is also important in maintaining protein homeostasis in the IMS [24]. Substrate specificities of the mitochondrial proteases vary considerably: For example, m- and i-AAA proteases degrade mainly membrane-embedded proteins [25], whereas LONP1–CLPP proteases degrade respiratory complex I to limit ROS production by depolarized mitochondria [26]. Upon oxidative damage, the TCA cycle enzyme Aconitase is specifically degraded by LONP1 protease in the matrix [27]. The single membrane spanning protein EMRE is degraded by m-AAA protease [28], whereas the turnover of lipid transfer proteins of the PRELID/Ups family in the IMS is mediated by YME1L [29]. The i-AAA protease YME1L also mediates turnover of several subunits of the TIM23 and TIM22 complexes, thereby regulating the mitochondrial import capacity according to the physiological demands of the cell [30]. By contrast, the PARL protease cleaves diverse substrates such as STARD7, PINK1, and PGAM5 within their transmembrane segments to regulate their membrane insertion and therefore subcellular localization [31]. It is noteworthy that the turnover rates of many of the mitochondrial proteins under normal conditions vary significantly. Nevertheless, at this time, how specific substrate proteins are selected for proteolysis by these mitoproteases is not well understood [31].

The ubiquitin-proteasome system (UPS) and the autophagy pathway serve as the principal protein quality control systems in the cell, yet their function extends to mitochondria by selectively eliminating damaged mitochondrial proteins or damaged mitochondria, respectively [32]. Both these pathways depend on specific ubiquitin signals to either selectively degrade an individual mitochondrial protein [33], or to completely eliminate a dysfunctional mitochondrion [34]. The UPS is a multistep and multicomponent process for intracellular protein degradation. A cascade of enzymes orchestrates selection, tagging, and degradation of targets through a series of steps: (i) ATP-dependent activation of ubiquitin by the E1 enzyme, (ii) conjugation of activated ubiquitin to one of the several E2s, (iii) transfer of ubiquitin to the target protein that is bound to a specific E3 ubiquitin ligase, and (iv) proteolysis of the conjugated target by the proteasome, an ATP-dependent multicatalytic protease, while releasing ubiquitin for tagging of subsequent substrates [35]. Invariably, all the diverse substrates of the UPS are polyubiquitinated and degraded by the proteasome. In this manner, the UPS regulates myriad cellular processes such as DNA replication, gene expression, cell cycle, and metabolism. As a consequence, impaired UPS has been implicated in human diseases from neurodegeneration to cancer [36].

As UPS components are generally localized in the cytosol, outer mitochondrial membrane proteins (OMM) such as MFN1, MFN2, TOM20 on the cytosolic side of mitochondria are readily recognized, ubiquitinated, and degraded by UPS (Figure 1). The ubiquitination of these proteins is mediated by certain E3 ubiquitin ligases that either constitutively associate with the OMM such as MITOL/MARCH5, or are recruited upon mitochondrial damage such as HUWE1 or Parkin [37,38,39]. On the contrary, the ROS produced by mitochondria under stressful conditions in yeast has been shown to regulate the NEDDylation and therefore stability of the SCF E3 ligase complexes [40], suggesting a crosstalk between mitochondria and UPS. Reminiscent of endoplasmic reticulum-associated degradation (ERAD), where UPS degrades ER proteins, a similar mechanism called mitochondria-associated degradation (MAD) exists for degradation of OMM proteins (Figure 1). It has been shown that like ERAD, Cdc48/p97 is recruited to the stressed mitochondria, extracts the ubiquitinated proteins, and subjects them to the proteasome for degradation [15,41,42].

Mitochondrial network dynamics adds another layer of mitochondrial quality control. Mitochondria constantly undergo fusion and fission which is necessary for cell survival, cell growth, and cell division. A complex machinery consisting of an elaborate set of proteins regulates the mitochondrial fusion and fission events. The fusion machinery constitutes multiple GTPases: MFN1, MFN2, and OPA1; whereas the fission machinery relies on a cytosolic GTPase DRP1 which interacts with receptor proteins on the OMM such as MFF, FIS-1, MID49, and MID51 [43,44]. Levels of many individual components of this machinery are regulated by selective ubiquitin-proteasome turnover [38,41,45,46]. Mitochondrial dynamics serve as a mechanism to preserve healthy mitochondria via fusion and eliminate the damaged ones by fission, subsequently subjecting them to removal by mitophagy [46]. In a separate mechanism for elimination of damaged mitochondria, the E3 ubiquitin ligase Parkin is activated by the PINK1 kinase upon loss of mitochondria membrane potential, and triggers mitophagy by massively ubiquitinating damaged mitochondria [47,48]. These ubiquitinated mitochondria are recognized by multiple autophagy-specific ubiquitin-receptors, which coordinate the assembly of autophagosomal membranes to engulf the damaged mitochondria in a process termed “ubiquitin-dependent mitophagy” (Figure 1). A related mode of mitochondrial quality control to eliminate mitochondria damaged by oxidative stress involves mitochondria-derived vesicles (MDVs). This pathway is also PINK1/Parkin-dependent, but unlike canonical mitophagy, it is triggered by ROS and not due to mitochondrial depolarization. The MDVs carrying oxidized mitochondrial proteins directly fuse with lysosomes independent of the autophagy pathway, suggesting that this mechanism is more efficient in clearing oxidized mitochondria without waiting for the mitochondrial depolarization to activate the canonical pathway of mitophagy [49,50]. 

In contrast to ubiquitination of outer membrane proteins, in this review we focus on newly discovered findings of intramitochondrial protein ubiquitination and their implications to mitochondrial protein import and metabolic regulation. 

## 2. Mechanisms of Mitochondrial Precursor Protein Ubiquitination and Degradation in Cytosol

Despite having their own genome, a vast majority of the mitochondrial proteins are encoded by the nuclear genome, translated by the cytosolic ribosomes and actively imported from the cytosol into mitochondria [51,52]. One of the earliest reports claiming turnover of inner membrane proteins by proteasome in the cytosol was the uncoupling protein UCP2, suggesting at the time that even the intramitochondrial proteins, could undergo ubiquitination [53]. Impaired mitochondrial protein import process leads to the accumulation of mitochondrial precursor proteins in the cytosol. In yeast, these mistargeted proteins bearing a hydrophobic tag trigger proteasome activation for their clearance through a pathway known as UPRam (unfolded protein response activated by mistargeting of proteins). This pathway is critical for protecting cells from protein import defect-related mitochondrial dysfunction and associated human pathologies (Figure 1) [54]. A related phenomenon named mPOS (mitochondrial precursor overaccumulation stress) in yeast is characterized by the accumulation of mitochondrial precursor proteins in the cytosol induced in response to defective mitochondrial import, compromised integrity of the IMM and mitochondrial dysfunction or damage. mPOS triggers expression of several ribosome-associated proteins that in turn suppress mPOS as part of a feedback loop mechanism. mPOS thus invariably links stress-associated mitochondrial dysfunction to cytosolic proteostatic stress (Figure 1) [55]. Although these pathways are induced in response to mitochondrial stress, they are an integral part of the cytosolic protein quality control mechanism. Indeed, proteasome inhibition led to the accumulation of many intramitochondrial proteins in a ubiquitinated state, suggesting that the UPS intimately regulates mitochondrial biogenesis even under basal conditions independent of mitochondrial stress [56].

A distinct subset of intramitochondrial proteins are targeted to the intermembrane space (IMS), and assembled by the mitochondrial intermembrane space import and assembly (MIA) pathway [57]. Substrates of MIA are small cysteine-rich proteins that need to form disulfide bonds in order to fold properly, thus the process of precursor translocation is tightly coupled to precursor oxidation. In yeast, some of these MIA substrates accumulate in the cytosol as polyubiquitinated precursor proteins following proteasomal inhibition, suggesting that a fraction of IMS proteins are degraded by the UPS prior to their import under basal conditions. Thus, the contribution of UPS to import of MIA substrates is not limited to import defects, but it constitutively removes a fraction of IMS precursor proteins under physiological conditions as a negative regulator during mitochondrial biogenesis [58]. Surprisingly, in yeast under conditions of defective protein folding (disulfide bond reduction), the conformationally destabilized mature IMS proteins are released from the mitochondria and cleared by proteasome, implying that a portion of IMS proteins may be retro-translocated to the cytosol for degradation by UPS [59]. A portion of these retro-translocated proteins may be ubiquitinated at mitochondria. In mammalian cells, a novel mechanism to regulate the import and quality of mitochondrial proteins has been attributed to a cytosolic protease DPP8/9. It prevents accumulation of mitochondrial precursor proteins in the cytosol by employing the N-end-rule degradation pathway. IMS proteins are the principal substrates of DPP8/9 protease that cleaves their N-terminus and the N-terminal amino group becomes acetylated by N-acetyl transferase to generate a degron that signals for proteasomal degradation [60].

Ubiquitination of proteins originating from, or targeted to, other mitochondrial compartments has also been documented. Thus, in mammalian cells, ubiquitin receptors from the Ubiquilin family have been shown to be critical in mediating degradation of mitochondrial transmembrane proteins if their import fails [61]. All these pathways eliminate unimported mitochondrial precursor proteins by ubiquitin-proteasome-dependent proteolysis in the cytosol as part of a general cellular response to proteotoxic stress. To what extent this reflects a distinct machinery from “cytosolic protein quality control”, whereby mistargeted or mislocalized proteins are identified by hydrophobic patches in the cytosol, ubiquitinated and degraded [62], or a dedicated pathway to regulate mitochondrial biogenesis and health, awaits to be resolved. 

## 3. Mechanisms of Intramitochondrial Protein Ubiquitination at Mitochondria

A pioneering study by Margineantu et al. on mammalian cells established a novel connection between the mitochondrial and cytosolic compartments through the ubiquitin-proteasome system, evident from post-transcriptional accumulation of mitochondrial proteins upon HSP90 inhibition [63]. Numerous proteins annotated as intramitochondrial are found in ubiquitinated state at mitochondria in yeast, most likely when they are exposed on the cytosolic surface of OMM. Among these are many proteins designated as residents of mitochondrial matrix, suggesting a critical role of UPS in regulating mitochondrial matrix proteins [64]. Occasionally, proteins may be stuck in the TOM complex during their import process and ubiquitinated as a result. These conjugates recruit proteasome and other UPS components to aid their removal and turnover as demonstrated in mammalian cells [56]. As mentioned above, mitochondrial protein import is a highly complex process which starts with recognition of the precursor protein by the primary presequence receptor TOM20, and continues by transfer onto the central receptor TOM22. The third receptor of the TOM complex, TOM70, is critical for import of noncleavable hydrophobic precursors. The recognition step is followed by translocation through the TOM40 channel, and continues via several different pathways depending on the final destination of the protein. This includes proteolytic cleavages, folding into large protein complexes, translocation through the TIM complex into the matrix, or insertion into the inner membrane [13,65,66]. Bearing the complexity of these processes in mind, it is expected that a portion of mitochondrial proteins may stall through the TOM complex, and clog the import channel. Defective mitochondrial protein import, therefore, leads to the accumulation of mitochondrial precursor proteins, which in turn can trigger diverse cellular stress response pathways. Recently, a number of mechanisms were identified as important for reducing accumulation of mitochondrial precursor proteins at the mitochondrial surface. These pathways continuously monitor the mitochondrial protein import channel TOM and mount a response that aids in selectively eliminating the proteins clogging the import channel and those incoming precursor proteins that accumulate at the mitochondrial surface awaiting their import.

A mechanism conserved from yeast to mammals, critical to circumvent mitochondrial stress promotes ubiquitin-dependent degradation of mitochondrial proteins and thus restores mitochondrial respiratory activity and cell viability. Specifically, in response to mitochondrial stress in yeast, the protein Vms1 translocate from cytosol to the mitochondria thereby aiding localization of Cdc48/VCP/p97 to the damaged mitochondria. Deletion of Vms1 leads to mitochondrial dysfunction, increased sensitivity to oxidative stress, and a corresponding reduction in life span, suggesting that Vms1-mediated mitochondrial surveillance by members of the UPS is critical for mitochondrial quality control and cell survival [42].

A pathway termed mitochondrial translocation associated degradation (mitoTAD) has been recently characterized in yeast which constantly monitors the TOM channel preventing its clogging by the incoming precursor proteins [67]. This mechanism ensures that mitochondria maintain their optimal protein import capacity under basal as well as stressful conditions (Figure 2). Mårtensson and coworkers identified a mitochondria-specific pool of the transmembrane ubiquitin-receptor Ubx2, distinct from the ER-resident pool of Ubx2 that is involved in ERAD. Ubx2 is characterized by a ubiquitin binding UBA (ubiquitin-associated) domain and a ubiquitin-like UBX (ubiquitin regulatory X) domain. Once imported into the mitochondria by the TOM70, Ubx2 exposes the UBA and UBX domains towards the cytosol. The UBX domain serves as the docking site for Cdc48 and its cofactors Ufd1-Npl4 whereas the UBA domain interacts with the substrate proteins (presumably trapped ubiquitinated precursor proteins from the TOM channel). How these targets are initially recognized and ubiquitinated at mitochondria remains to be elucidated. 

Distinct from the mitoTAD pathway, which constitutively monitors the mitochondria for clogged TOM channels, a stress-specific pathway that senses import defects has been identified in yeast known as mitochondrial compromised protein import response (mitoCPR) [68]. To induce mitochondrial import stress, Weidberg and coauthors overexpressed certain mitochondrial proteins encoding a bipartite signal sequence, thereby overloading the TIM23 translocase (rather than depolarizing the mitochondria membrane potential with the chemical CCCP, as is common in many experimental systems). The resulting defective mitochondrial protein import induced the expression of a set of genes, including CIS1, mediated by the transcription factor PDR3. Upon its expression, CIS1 localized to OMM through its interaction with TOM70 and recruits the AAA ATPase MSP1 to the TOM complex [68]. MSP1 is thus positioned to extract the mitochondrial precursor proteins from the TOM translocase, remove them from the mitochondrial surface, and subject them to proteasome for degradation (Figure 2). The mitoCPR pathway is thus mounted only under conditions of mitochondrial import defect, and serves to preserve mitochondrial function. Although the pathways discussed in this section mention proteasome-mediated proteolysis of accumulated mitochondrial precursor proteins that failed to enter mitochondria due to import defects, their mechanism of ubiquitination or the specific E3 ubiquitin ligases that carry it out have not yet been identified. 

## 4. Mitochondrial Import-Associated Ubiquitination

The UPS plays a dual role in mitochondrial protein import. Indirectly, ubiquitination contributes by unclogging blocked import channels, but deubiquitination can also directly promote the import process. Very recently, Phu et al. demonstrated in mammalian cells that the OMM resident E3 ligase, MARCH5, constitutively ubiquitinates the incoming mitochondrial precursor proteins, thereby inhibiting their import [69]. This was evident from the fact that silencing MARCH5 reduced the extent of ubiquitination of intramitochondrial proteins, whereas overexpression of MARCH5 led to the accumulation of ubiquitinated intermediate and precursor forms and a concomitant reduction in the mature form of the proteins. Counteracting MARCH5, the OMM resident deubiquitinase USP30 deubiquitinate the incoming ubiquitinated precursor proteins at the TOM translocase, which aids their import. Abolishing USP30 activity caused these ubiquitinated precursors to be fed to the proteasome. Thus, genetic deletion of USP30 led to the accumulation of ubiquitinated import intermediates at the surface of mitochondria when proteasome was inhibited, suggesting that USP30 activity at the TOM is critical for import. The authors further deduced that MARCH5 overexpression promotes ubiquitination of the precursor proteins via K48, K11, and K63 linkages predominantly in cells lacking USP30 [69]. The association of USP30 with the TOM complex has also been confirmed by Ordureau and colleagues. The ubiquitylation of several TOM complex subunits increased in USP30 knockout neurons under basal conditions as well as in response to mitochondrial membrane depolarization, suggesting that even the mitochondrial import channels are regulated by ubiquitination to fine tune the mitochondrial protein import [70]. Interestingly, ubiquitination of IMS proteins in yeast prevents their mitochondrial import, further implying that deubiquitination of the mitochondrial precursor proteins is critical for their import [71]. UPS thus play a major role in mitochondrial biogenesis and regulates the composition of mitochondrial proteome by regulating protein import into mitochondria. Another important aspect of this type of ubiquitination mechanism is to add a layer of quality control over the precursor protein that persists in an unfolded extended polypeptide state prone to misfolding or aggregation during the import process despite being protected by the chaperones. This exciting finding unveils a novel role of ubiquitination and deubiquitination in the regulation of mitochondrial protein import (Figure 2). 

## 5. Mechanisms of Mitochondrial Export-Associated Ubiquitination

The TOM40 channel allows passage only of unfolded proteins during their import into the mitochondria. Thus, theoretically it may also allow the export of damaged unfolded proteins from the mitochondria back to the cytosol for degradation, a process termed as retrotranslocation. This possibility was evaluated by Bragoszewski et al. and showed that in yeast, the IMS proteins, which are imported by the MIA pathway, can be exported from the IMS following their inability to fold [59]. These IMS proteins depend on their intrinsic Cysteine residues to form disulfide bonds in order to fold properly (termed as oxidative folding) which is also critical for them to localize in the IMS (called as oxidative folding trap). The authors demonstrated that reduction of disulfide bonds in IMS proteins prevented their folding, causing them to escape through the TOM40 channel. Since the exported (retrotranslocated) proteins were unfolded, they were readily ubiquitinated and degraded by the proteasome [59]. The authors later suggested that a portion of these retro-translocated polypeptides may be ubiquitinated at the outer surface of the mitochondria, presumably as they exit the TOM channel [72] (Figure 2). Whether this pathway is conserved, at least partially, in the mammalian cells is largely unknown. Nevertheless, like mitoTAD, the existence of a similar mechanism in higher eukaryotes, constitutively operational to degrade any misfolded proteins through retrotranslocation, warrants further research. 

## 6. Implications of Intramitochondrial Protein Ubiquitination: Metabolic Regulation by UPS

Mitochondria have long been known to attain different morphologies in diverse cell and tissue types and shaped by the extrinsic signals and metabolic requirements of the cells, a characteristic property known as “mitochondrial plasticity” [73]. Structural diversity of mitochondria arising from dynamics of the mitochondrial inner and outer membrane finds implications in a variety of cellular processes [74]. However, by far the most critical role of mitochondrial dynamics has been underscored in the regulation of mitochondrial bioenergetics [75,76,77,78]. An elaborate set of proteins localizing to both the mitochondrial inner and outer membrane regulates dynamics of the mitochondrial network and in turn bioenergetics in both yeast and mammalian cells [74,79]. As mentioned above, the turnover of many of these proteins is regulated by UPS [33,38,45]. Thus, it is apparent that UPS regulates mitochondrial bioenergetics indirectly through modulation of mitochondrial network dynamics. 

Regulation of cellular metabolism by the UPS has been a subject of investigation over several decades since the identification of the UPS itself. Ever since, many key cellular metabolic pathways, including the AMPK pathway, have been found to be modulated by the UPS [80,81]. The contribution of the UPS to mitochondrial metabolic regulation is just beginning to be revealed and already a surprisingly large number of mitochondrial metabolic enzymes have been shown to be regulated by the UPS. Mitochondria are metabolic hubs coordinating between respiration (oxidative phosphorylation), TCA cycle, βoxidation, amino acid metabolism, and other processes which require tight regulation in order to maintain ATP supply in changing environmental/growth conditions. Like most other intramitochondrial proteins, a majority of the enzymes involved in these mitochondrial metabolic pathways are nuclear encoded, imported from cytosol and in their mature forms, are inaccessible for direct regulation by the cytosolic UPS. Hypoxia is a common stress in cells of solid tumors and causes a metabolic shift from oxygen dependent to glycolytic pathway. α-ketoglutarate dehydrogenase (αKGDH) is an important mitochondrial enzyme complex involved in glutamine oxidation and TCA cycle. In the absence of αKGDH, glutamine follows a reductive pathway that produces excess citrate and promotes lipid synthesis. The enzyme oxoglutarate dehydrogenase (OGDH) is a subunit of the αKGDH complex that is downregulated in hypoxia. The 48kD isoform of OGDH was found to be ubiquitinated by the cytosolic E3 ligase SIAH2 under hypoxia, resulting in proteasomal degradation [82]. This UPS dependent degradation of a mitochondrial matrix enzyme facilitates the metabolic shift from glutamine oxidation to the glutamine reduction pathway. Whether the ubiquitination of OGDH happens prior to its import into mitochondria, during import, or after export remains to be determined. Interestingly, the deubiquitinating enzyme USP13 was found to regulate the levels of a different 115kD isoform of OGDH in ovarian cancer (OVCA). USP13 directly deubiquitinates OGDH, and its amplification in OVCA favors glutamine reduction and lipid metabolism [83].

An example of direct evidence of UPS-mediated regulation of mitochondrial metabolism emerged recently with the critical role of UPS in metabolic shift depending on availability of the cellular energy source. Complex II (Succinate Dehydrogenase-SDH) of the ETC is an IMM enzymatic complex that catalyzes the dehydrogenation of succinate to fumarate. When glucose or glutamine was available as energy sources, the turnover of SDHA (Succinate Dehydrogenase Subunit A) was regulated by the UPS. However, when palmitoyl-D, L-carnitine served as the energy source to produce energy via β-oxidation, the SDHA levels were highly stable and not degraded by the UPS. This corresponds to the fact that the SDH complex is more active under β-oxidation compared with pyruvate during TCA cycle [84]. A clinical manifestation of a defective UPS component and its relevance to mitopathy is highlighted by FBXL4. The F-box protein FBXL4, a substrate-adaptor component in the Skp1-cullin1-F-box (SCF) E3 ubiquitin ligase, is critical for mitochondrial quality control. FBXL4 defects are associated with congenital lactic acidemia and encephalomyopathic mitochondrial DNA depletion syndrome [85]. FBXL4 deficiency causes increased turnover of mitochondria through autophagy, leading to an overall decrease in mitochondrial content, a reduced OXPHOS activity, and compromised mitochondrial metabolic activity [86,87]. These observations suggest that even the turnover of mitochondrial metabolic enzymes can be fine-tuned by the nominally cytosolic UPS to adapt to the metabolic requirements of the cells depending on the availability of specific substrates as energy source.

The plausible explanation for the ubiquitination of these mitochondrial metabolic enzymes is that they are synthesized in excess and degraded as ubiquitinated precursor proteins in the cytosol (by mechanisms detailed above). Another explanation is that their mitochondrial abundance is tightly regulated by import-associated ubiquitination and deubiquitination (see earlier section) in a metabolic-state associated manner. We cannot rule out the possibility of unfolding and export of damaged intramitochondrial proteins and retro-translocation to the cytosol contributing to mitochondrial metabolism as well. 

## 7. Turnover of Respiratory Chain Subunits by UPS

The subunits of the electron transport chain complexes constitute a substantial proportion of intramitochondrial proteins. A majority of these subunits are encoded by the nuclear genome, synthesized in the cytosol, and have to be imported inside the mitochondria. Moreover, these imported subunits have to be assembled together with those components that are synthesized within the mitochondria to form functional complexes (respirasomes). The import and assembly of these subunits has recently been shown to be closely monitored by the UPS, evident from the fact that they are constantly ubiquitinated even under basal conditions. A large-scale proteomic study of the mitochondrial proteome in *Drosophila melanogaster* brains revealed that the turnover rate of many respiratory chain (RC) proteins is significantly compromised in PINK1 and Parkin-mutant flies as compared to the Atg7 mutant (autophagy deficient) flies, suggesting that UPS is critical in turnover of these proteins even under basal conditions [88]. SILAC proteomics approach revealed differential assembly rates of various RC complexes which reflects their functional efficiencies, for instance, the assembly of RCV (ATP synthase) is rapid but that of RCI (NADH Dehydrogenase) is relatively less efficient. The excess unassembled subunits are usually degraded rapidly [89], suggesting that import, folding, and incorporation into quaternary structures of key subunits may be rate limiting relative to others. Indeed, a recent proteomic screen of the mitochondrial proteome by di-glycine remnant approach in response to proteasome inhibition revealed accumulation of many OXPHOS subunits and metabolic enzymes, suggesting their regulation by UPS under basal conditions [56]. The regulation of OXPHOS subunits through ubiquitination thus finds implications in their proper import, folding, and assembly into functional supercomplexes.

## 8. Concluding Remarks

Mitochondria and the ubiquitin proteasome system directly influence each other [90]. The role of UPS in maintenance of healthy mitochondria has been classically limited to stressful conditions and restricted to OMM proteins, due to their accessibility. In the past three years, the extent of the UPS effect has transpired to encompass all aspects of mitochondria morphology and physiology. Recent reports provide evidence that UPS can even regulate intramitochondrial proteins and control aspects of protein import, metabolism, and respiration under basal conditions. An image emerges of mitochondria that are constantly under surveillance of the cytosolic UPS, with regulation of intramitochondrial proteins by the UPS providing a survival advantage to the cells by optimally adapting to the growth conditions. In the endosymbiotic relationship of mitochondria and eukaryotic cells, mitochondria provide ATP and various metabolites to their host cells (as well as occasional toxins) [91]. The host cell in return provides UPS-mediated quality control surveillance to the mitochondria. It is now clear that the host cell influences all aspects of the mitochondrion proteome, and by extension, also its metabolism. It is truly a two-way relationship.

## Figures and Tables

**Figure 1 biomolecules-10-01559-f001:**
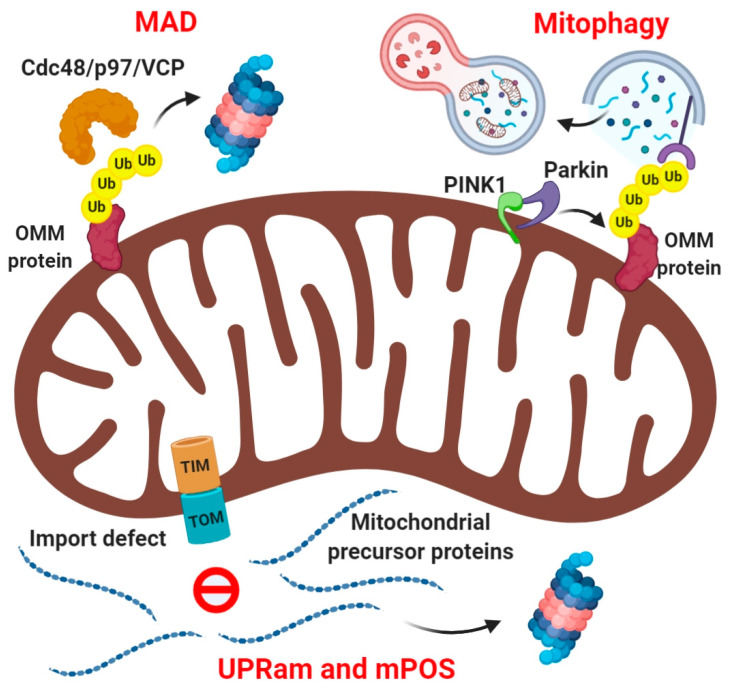
Regulation of mitochondrial protein quality control by cytosolic ubiquitin-proteasome system (UPS). Individually damaged proteins in the outer mitochondrial membrane (OMM), are selectively ubiquitinated by specific E3 ligases, extracted by the AAA ATPase Cdc48/VCP/p97 and then degraded by the proteasome by “MAD” pathway. Irreversible damage to the mitochondria leads to PINK1 stabilization at OMM and subsequent recruitment and activation of E3 ligase Parkin, which extensively ubiquitinates the OMM proteins. These ubiquitin chains serve as signals for engulfment of damaged mitochondria by autophagosomes, through the pathway of “ubiquitin-dependent mitophagy”. Impaired mitochondrial protein import and sorting pathways leads to the accumulation of mitochondrial precursor proteins in the cytosol, which triggers their clearance by the proteasome through pathways known as UPRam and mPOS.

**Figure 2 biomolecules-10-01559-f002:**
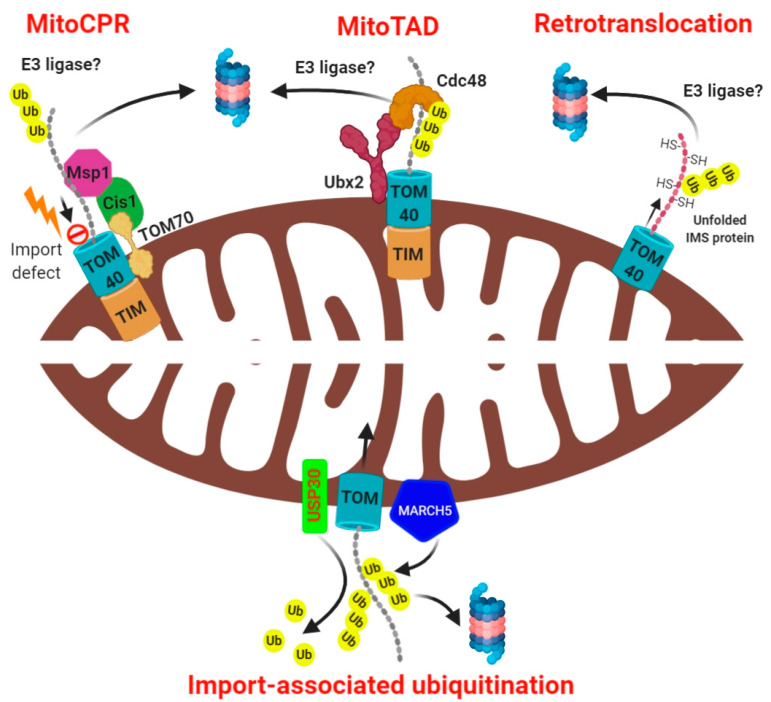
Modes of mitochondria-associated ubiquitination and degradation. The mitoCPR pathway entails that stress-induced impaired mitochondrial protein import leads to localization of a protein Cis1 at the TOM complex through its interaction with TOM70. Cis1 further recruits the AAA ATPase Msp1 at the TOM complex which extracts the ubiquitinated protein and subject it to the proteasome. The “MitoTAD” pathway constitutively performs surveillance of the import channel, clearing any blockade through association of a protein Ubx2 with TOM. The retrotranslocation pathway involves export of unfolded intramitochondrial proteins to the cytosol for their ubiquitin-dependent proteasomal degradation. The OMM resident E3 ligase MARCH5 constitutively ubiquitinates the incoming precursor proteins whereas the deubiquitinase USP30 removes the ubiquitin, thereby facilitating the mitochondrial protein import.

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
