# Peer review of "Ubiquitination of Intramitochondrial Proteins: Implications for Metabolic Adaptability"

_biomolecules, 2020, doi:10.3390/biom10111559_

Round 1
Reviewer 1 Report
The review article by Sukshane and colleagues summarizes recent findings about the role of the ubiquitin proteasome system for the degradation of intramitochondrial proteins. Sukshane and colleagues provide a comprehensive overview about a rapidly developing scientific field. The reviews highlight all relevant findings and elegantly combines them with open issues discussed in the field. I have a few recommendations as listed below for the revision of the manuscript. After revision of the points listed below, I support publication of this manuscript in Biomolecules.
In general, the authors should clearly mention the corresponding organism, in which the finding was made. In the current version, this is missing at some positions.
Line 70: Replace TIM complex with TIM23 complex. There are two inner membrane TIM complexes. In order to avoid confusion, the authors should name the protein translocases that is required here.
Lines 75 pp.: The authors should mention that most proteins that engage the mitochondrial Hsp70 are sorted into the mitochondrial matrix. In this context, it should be noted that the mitochondrial Hsp70 functions in protein import at the TIM23 complex and as protein folder in the matrix.
Lines 82 pp.: The authors should name some substrates of the mitochondrial proteases to substantiate the different substrate specificities of the proteases.
Lines 99 pp.: The authors should describe that many substrates of the ubiquitin-proteasome system are polyubiquitinated.
Lines 183 pp.: The authors should describe the conditions under which mature intermembrane space proteins are degraded by the proteasome.
Line 210: The authors should state that TOM20 and TOM70 recognize incoming precursor proteins and not only TOM20 as stated in the text.
Lines 237 pp.: The UBX domain and not the UBA domain recruit CDC48 to the TOM complex. This should be corrected.
Figure 2: The TOM complex should be in the outer membrane. In the current version, it appears that the TOM complex is in the cytosol.
Lines 272 pp.: The description about the role ubiquitination of precursor proteins in the import into mammalian mitochondria should corrected. Ubiquitination impairs protein import, but does not stimulate protein import as stated in the text. Deubiquitnation promotes import into mitochondria. In the context, it worth mentioning that Chancinska and colleagues found that ubiquitinated proteins cannot pass the TOM channel (Kowalski et al., 2018). Furthermore, it should be mentioned that these findings are made in mammalian, but not in yeast. Finally, the authors should include the paper by Ordureau and colleagues in Mol Cell 2020 here as they found a role of USP30 at the TOM complex.
The authors should include the uncoupling proteins as further examples of inner membrane proteins that are degraded by the ubiquitin proteasome system.
Author Response
- In general, the authors should clearly mention the corresponding organism, in which the finding was made. In the current version, this is missing at some positions.
Re: The corresponding organism has now been mentioned in the respective positions in the revised version (“Organism mentioned” in the comments section).
- Line 70: Replace TIM complex with TIM23 complex. There are two inner membrane TIM complexes. In order to avoid confusion, the authors should name the protein translocases that is required here.
Re: TIM complex replaced with TIM23 complex (Line 70).
- Lines 75 pp.: The authors should mention that most proteins that engage the mitochondrial Hsp70 are sorted into the mitochondrial matrix. In this context, it should be noted that the mitochondrial Hsp70 functions in protein import at the TIM23 complex and as protein folder in the matrix.
Re: We included the sentence “Most proteins that engage the mitochondrial HSP70 are sorted into the mitochondrial matrix and thus HSP70 facilitate protein import at the TIM23 channel by functioning as protein folder in the matrix” as per the suggestion of reviewer 1 (Lines 75-77).
- Lines 82 pp.: The authors should name some substrates of the mitochondrial proteases to substantiate the different substrate specificities of the proteases.
Re: We enlist representative mitochondrial proteases with their substrate specificities (Lines 95-108).
- Lines 99 pp.: The authors should describe that many substrates of the ubiquitin-proteasome system are polyubiquitinated.
Re: We mention that “Invariably, all the diverse substrates of the UPS are polyubiquitinated and degraded by the proteasome” (Line 120).
- Lines 183 pp.: The authors should describe the conditions under which mature intermembrane space proteins are degraded by the proteasome.
Re: We mention here that “In yeast under conditions of defective protein folding (disulfide bond reduction), the conformationally destabilized mature IMS proteins are released from the mitochondria and cleared by proteasome” (Line 201-203).
- Line 210: The authors should state that TOM20 and TOM70 recognize incoming precursor proteins and not only TOM20 as stated in the text.
Re: We emphasize here the importance of TOM70 receptor during mitochondrial import. “The third receptor of the TOM complex, TOM70 is critical in import of noncleavable hydrophobic precursors.” (Line 233-235).
- Lines 237 pp.: The UBX domain and not the UBA domain recruit CDC48 to the TOM complex. This should be corrected.
Re: We made a correction here that UBX domain and not the UBA domain recruit CDC48 to the TOM complex. “Once imported into the mitochondria by the TOM70, Ubx2 exposes the UBA and UBX domains towards the cytosol. The UBX domain serve as the docking site for Cdc48 and its cofactors Ufd1-Npl4 whereas the UBA domain interacts with the substrate proteins (presumably trapped ubiquitinated precursor proteins from the TOM channel)” (Line 260-265).
- Figure 2: The TOM complex should be in the outer membrane. In the current version, it appears that the TOM complex is in the cytosol.
Re: We made the necessary changes in both the figures.
- Lines 272 pp.: The description about the role ubiquitination of precursor proteins in the import into mammalian mitochondria should corrected. Ubiquitination impairs protein import, but does not stimulate protein import as stated in the text. Deubiquitnation promotes import into mitochondria. In the context, it worth mentioning that Chancinska and colleagues found that ubiquitinated proteins cannot pass the TOM channel (Kowalski et al., 2018). Furthermore, it should be mentioned that these findings are made in mammalian, but not in yeast. Finally, the authors should include the paper by Ordureau and colleagues in Mol Cell 2020 here as they found a role of USP30 at the TOM complex.
Re:
- We removed the discussion here about “MARCH5-mediated ubiquitination stimulates protein import” (Line 319-321).
- We mention here work by Ordureau and colleagues that shows importance of USP30 in deubiquitylation of TOM complex subunits (Line 314-318).
- We also include the work by Chancinska and colleagues which demonstrate that ubiquitinated proteins cannot pass through the TOM complex (Line 321-323).
- The authors should include the uncoupling proteins as further examples of inner membrane proteins that are degraded by the ubiquitin proteasome system.
Re: We mention here that the IMM uncoupling protein UCP2 was one of the earliest discoveries that IMM proteins are degraded by the UPS (Line 174-176).
Reviewer 2 Report
In this manuscript, the authors review the ubiquitination of intramitochondrial proteins and its implications for metabolic adaptability. Metabolic homeostasis is one of the main mitochondrial function, and despite growing evidences supporting a role of UPS in the regulation of this metabolism (e.g. Lavie J. et al cell reports 2018, Bonnen PE et al Am. Jour. Hum Genet. 2013), no review article has addressed this question. Indeed, most of reviews address links between mitochondria and UPS question through the Parkin/mitophagy prism and impacts on mitochondrial physiology is broadly underestimated.
This article is well organized and provide all the keys to understand this important question. I would suggest to include some discussion regarding the E3 ubiquitine ligase, FBXL4, which is responsible for a severe mitochondrial disease characterized by alterations of mitochondrial metabolism.
To me, this review provides an exhaustive view of the topic including some important studies (Livnat-Levanon et al; Lavie et al etc…). Yet, I would also include the pioneering work of Margineantu, D.H., et al PLoS ONE 2007 should be include as it revealed ubiquitination of intramitochondrial proteins.
Provided the authors addressed these minor concerns, this review article strongly deserves to be published.
Author Response
- I would suggest to include some discussion regarding the E3 ubiquitine ligase, FBXL4, which is responsible for a severe mitochondrial disease characterized by alterations of mitochondrial metabolism.
Re: We summarize here the role of FBXL4 in mitochondrial quality control and how its deficiency leads to mitopathy (Line 391-397).
- I would also include the pioneering work of Margineantu, D.H., et al PLoS ONE 2007 should be include as it revealed ubiquitination of intramitochondrial proteins.
Re: We include here the work by Margineantu et al. which for the first time showed a direct connection between mitochondria and cytosolic UPS (Line 222-225).